# Photons for Photography: A First Diagnostic Approach to Polaroid Emulsion Transfer on Paper in Paolo Gioli’s Artworks

**DOI:** 10.3390/molecules27207023

**Published:** 2022-10-18

**Authors:** Zeynep Alp, Alessandro Ciccola, Ilaria Serafini, Alessandro Nucara, Paolo Postorino, Alessandra Gentili, Roberta Curini, Gabriele Favero

**Affiliations:** 1Department of Environmental Biology, Sapienza University of Rome, 00185 Rome, Italy; 2Department of Chemistry, Sapienza University of Rome, 00185 Rome, Italy; 3Department of Physics, Sapienza University of Rome, 00185 Rome, Italy

**Keywords:** polaroid chemistry, polacolor, fibre optics reflectance spectroscopy, Paolo Gioli, chromium (III) complex of azomethine dye, chromium complex of an azo pyrazolone, blue copper phthalocyanine

## Abstract

The aim of this research is to study and diagnose for the first time the Polaroid emulsion transfer in the contemporary artist Paolo Gioli’s artworks to provide preliminary knowledge about the materials of his artworks and the appropriate protocols which can be applied for future studies. The spectral analysis performed followed a multi-technical approach first on the mock-up samples created following Gioli’s technique and on one original artwork of Gioli, composed by: FORS (Fiber Optics Reflectance), Raman, and FTIR (Fourier-Transform InfraRed) spectroscopies. These techniques were chosen according to their completely non-invasiveness and no requirement for sample collection. The obtained spectra from FTIR were not sufficient to assign the dyes found in the transferred Polaroid emulsion. However, they provided significant information about the cellulose-based materials. The most diagnostic results were obtained from FORS for the determination of the dye developers present in the mock-up sample which was obtained from Polacolor Type 88 and from Paolo Gioli’s original artwork created with Polacolor type 89.

## 1. Introduction

Conservation of contemporary artworks is a new thrilling field that needs continuous research due to the utilization of the newest and ever-changing materials, mediums, and ideologies in the making of a work of art. Contemporary art considers the use of all materials, along with different forms of exposition, ranging from the absence of material to its accumulation. A restorer must pay good attention to the composition of the artwork and consecutively to the decision-making process regarding the conservation steps which will be applied to the artwork. At this point, the detailed knowledge about the artwork materials on a scientific level is crucial. Therefore, what conservation science provides to the literature through analytical techniques is of great importance for art preservation. From this perspective, this study presents research diagnosing Paolo Gioli’s Polaroid emulsion transfers by following a multi-spectral approach. Gioli was an Italian contemporary artist born in Sarzano (Rovigo) on 12 October 1942. The above-mentioned significance of the artist’s material plays even a bigger role for Gioli’s art because he used the Polaroid for its material plasticity and versatility. He transferred the emulsion to different receptor layers (paper, silk, wood, etc.), while denying having used fully what the Polaroid Corporation has to offer, from the cameras to the instructions. The artworks he created are of great complexity on an aesthetic level and, also, on a chemical one. Since knowledge about the chemistry and the conservation of Polaroid materials is not broad enough, analysing the Polaroid materials transferred to a different support is an important contribution and a discovery of a different aspect of the photographic materials from the scientific point of view. It is fundamental to mention that the original photographic material on which the analysis will be applied is aesthetically complete and in very good condition, and it is a perfect representation of the artists’ intention: every single trace contributes to the wholeness of the artwork. Therefore, also material-wise, the sampling from the material surface was impossible. For this reason, a completely non-invasive spectroscopic protocol was chosen both for the mock-up sample, produced by using the film Polacolor type 88, and the original artwork of Paolo Gioli, in which he used Polacolor type 89. Photographic material is expected to give complications while performing non-invasive diagnostics, due to its organic components, and this is a limitation of this research. Moreover, since no prior diagnostics has been carried out on Polaroid materials transferred on different receptor layers, in our case paper, it is hard to foresee the information potential achievable from single techniques. However, for the same reason, this represents the first ever research to provide primary information about the protocols to be followed for the analysis of such materials in the future. From the conservation science point of view, Polaroid film represents an undiscovered field and needs further research to fully determine the substances used. Paolo Gioli’s works being constructed of complex experimental layers of materials add a further dimension to future research.

Chemistry of photographic materials has been studied extensively by past scholars [1,2,3,4,5,6,7,8,9,10,11,12,13,14,15,16,17,18,19,20,21,22,23,24,25,26,27] through several analytical techniques [4,5,26,28,29,30,31,32,33,34,35,36,37,38,39,40,41,42,43,44]. The colour photography mechanism is based on silver halides (AgX) just as black and white photography. The silver halides in colour photography act as mediators for transforming light into organic dye images [15]. The use of colour filters for the colour photography dates to the production of colour images by James Clerk Maxwell in 1861, which relies on the additive colour system whereby the three primary colours (red, green, and blue) produce the full gamut of colours on the composite final image. On the other hand, the subtractive colour process, subtracts red, green, or blue light from the visible spectrum. Therefore, the original image produces a positive, through the intermediate step of a negative image. Cyan, magenta, and yellow are the colours used for the subtractive colour system [23].

The mechanism of instant photography was first published in 1947 [45]. The Polaroid Corporation published the early instant photography concepts in 1948 of black and white [46]. The beginning of the coloured peel-apart system was put on the market in 1963 by Polaroid Corporation, Cambridge, NA, USA Afterwards, integrated (mono-sheet) systems called SX-70 (1972, Polaroid), PR-10 (1976, Kodak, New York, NY, USA), and FI-10 (1981, Fuji Photo Film, Tokyo, Japan) appeared as more elaborate systems [15]. Polaroid prints film has the same subtractive colour principle as regular colour negative films. The main difference is that all the chemicals, including the developer and the dyes (referred as the developer-dye) are enclosed within a thin sheet of film. In the so-called “peel-apart” films, the negative and the positive is stripped apart to reveal the image [21]. 

A reagent system is an essential part of each instant film. Dry processing is realised by using a highly viscous gel reagent and therefore restricting the amount enough to complete a single photo. The viscosity of the reagent is provided by water-soluble polymeric thickeners including hydroxyethyl cellulose, the alkali metal salts of carboxymethyl cellulose, and carboxymethyl hydroxyethyl cellulose. The high viscosity enables the accurate metering to form a thin film and serves as an adhesive for the two sheets during processing. The reagent is highly alkaline, and this property can remain stable due to the sealed pod that contains the reagent. Sealing also protects the reagent from oxygen until the development of the film. The reagent contains reactive components that participate in image formation, deposition, and stabilisation [47]. The function of the developing agent consists in reducing exposed silver halide grains to metallic silver and thus facilitating the formation of the oxidized species of the developing agent. This species reacts with a colour coupler in the emulsion to form a dye [48].

In the Polacolor instant film, a single reagent is used to obtain a negative and a positive image. The quantity of reagent that is available to produce the positive image is controlled as a function of the development of the silver halide latent image as an oxidation product of the reagent, which is a substance that is immobile in a photosensitive element while depositing the unreached reagent on a print-receiving material to provide the positive image. The reagent, when oxidised, provides a reaction product which has a low solubility in a processing liquid than the unreached reagent [23]. In particular, the multilayer negative contains a set of blue-, green-, and red-sensitive emulsions. Each layer contains respectively a dye developer complementary in colour to the emulsion’s spectral sensitivity. A negative section comprises the lower part of the film, and a positive section the upper part. These sections are physically one unit until they become separated by the viscous alkaline reagent after exposure. The alkaline reagent is enclosed in pods at the edge of the film. It is spread between the two sections by automatic rollers [48]. The dye developers get oxidated and immobilised due to the development of the exposed grains during processing. The non-immobilised dye developer then migrates through the layers of the negative to the image-receiving surface to form the positive image. Dye developers must have good diffusion properties and must stay inert within the negative before that it is processed. The diffusion must also remain stable to light and have suitable spectral absorption characteristics after the processing [10].

When Polaroid Corporation first introduced Polacolor film in 1963, the dye developers were a set of two azo dyes for yellow and magenta and anthraquinone for cyan [23]. However, it was seen that over time these dyes needed further light fastness and stability. More recent developments in dye-developer chemistry have primarily focused on achieving further light fastness and stability and have resulted in the introduction of premetallized dyes in the Polacolor 2 and SX-70 material. The technique used for metallized dye images used the image wise transfer of dye developers which included chelating dye systems [49], especially metal complex azo dyes, renowned for their great stability to light [16,50,51,52,53].

## 2. Results and Discussion

### 2.1. Mock-Up

#### 2.1.1. FORS

Out of 20 points where acquisions were made, points T2, T5, T8, T14, T10, and T20 show the most intense bands (Figure 1), which result useful for the identification of the dyes. Bands centered around 490–505, 562–565, and 681–684 nm were the most recurring, as shown in Table 1.

The paper substrate does not affect or distort the obtained results because the colourant is abundant enough to absorb the light and, therefore, does not make it possible for the light to arrive to the paper surface. This has been proved by obtaining the apparent absorbance spectrum of the paper and comparing it with the absorbance spectra of the analogues of the coloured areas. Therefore, the reflectance of the paper has been ignored for the interpretation of the results obtained. 

For the identification of the dyes, we compared the experimental data with the literature characteristic absorption ranges and maxima for several dyes used in Polaroid films. In general, yellow, magenta, and cyan dye couplers ideally absorb the wavelength ranges specifically: 600–700 nm is the typical range for a cyan dye, 500–600 nm for a magenta dye, and 400–500 nm for a yellow dye.

In our case, as shown in Table 1, the absorption density around 682–685 nm is present for every analysed point, and it is associated with the absorption of the red component of visible light by the cyan dye. From research made on copper phthalocyanine dyes in solution [52,53,54,55], the absorption maxima of such dyes result at 678–688 nm, which are compatible to our results obtained from every sample point [15,24,48,49] (Figure 2).

The maxima around 560–565 nm wavelength range are also present in all the spectra; this is indicative of the magenta dye. For what is reported about Polaroid films, magenta dyes have azo-pyrazolone structure in coordination with Chromium, with absorption maxima varying also on the substituent groups of the dye [49] (Figure 3). In particular, magenta dye developer used for Polacor 2 films—with the X group being SNO_2_(CH_2_CH_2_OH)_2_, the Y group being C_6_H_5_, and the Z group being CH_3_—presents absorption maximum in the mentioned range, according to [49]. Therefore, this is likely the structure of the dye developer present in the mock-up (Figure 4). 

Regarding the yellow dye, it is important to notice that only the point T5 presents an evident maximum at 443 nm, while points T4, T6, T7, T9, T11, T12, T13, T14, T15, and T18 have maxima shifted around 500–505 nm. The other points do not show evident maxima in this range (Table 1), even if for all the spectra broad absorptions are observable in the extended 430–500 nm range. The research of [32] demonstrates that, due to degradation, the absorption maxima at around 440 nm of the yellow dye shift towards lower wavelengths, which in our case is the least detectable and least stable zone. Moreover, the bands tend to become wider and lower in intensity. Finally, yellow dyes could have some unwanted green absorption due to degradation phenomena [48]. In order to highlight which were the points which could be indicative of the original yellow dye and which could be more affected by degradation processes, colorimetric data (L* a* b* coordinates) were also obtained to understand the correlation of the absorbance results and the state of the yellow dye (Table 2). 

In the spectra corresponding to points T2, T5, and T20, a wide curve is observable between 409 and 480 nm (Figure 1), even if its remarkably broader in comparison to the band of other dyes (for instance, in the case of T2, the maximum cannot be identified with certainty). Moreover, no evident band is observable around 500 nm. The colorimetric data for these points presented the highest value for the b* coordinate, confirming a higher concentration for the yellow dye, while no particular evidence of degradation products involving a greenish shade is observable. On the other side, the point T9 corresponds to the lowest value of b* coordinate, while in the apparent absorbance spectrum a maximum of T9 at 500 nm is remarkable (see Appendix A). At this point, the degradation of the yellow dye to green degradation products can be hypothesized. A similar trend is observable for the other points. From these data, it is possible to affirm that, for Polaroid emulsions, bands between 400 and 490 nm are indicative of original yellow dyes, but their eventual broadness could not allow a specifical attribution, while signals at 500 nm are likely indicative of green degradation products, with similarities to phenomena observed for other typologies of photographic films [48].

#### 2.1.2. FTIR

Reflectance FTIR spectra present some drawbacks for the spectral interpretation; for example, since the sent light is being reflected, depending on the physical state of the material of interest, the results may show shifted, noisy, or negative peaks where there is supposed to be a maximum. This factor was considered while interpreting the results.

The FTIR spectra corresponding to the area where the emulsion transfer was made and the paper itself showed a similar pattern due to the gel state and the thinness of the measured film (Figure 5). The signals at 1088, 1129, 1370, and 1641 cm^−1^ and the broad absorption band between 3443 and 3715 cm^−1^ are characteristics of cellulose found in the paper [56], while the peaks at 1377 and 1433 cm^−1^ may suggest the presence of Arabic gum, used as binder of the cellulosic paper. Point F3, the lightest area in hue, shows a more similar spectrum to the paper than point F1 and F2. 

F1 differentiates from F3 and the paper from the fact that the peaks at 1088 and 1177 cm^−1^ in F1 are way less intense; meanwhile, the peak, found in F3 and paper at 1129 cm^−1^, is shifted to 1145 cm^−1^ in F1 and F2. The peak at 1281 cm^−1^ was found in F3, and the paper does not exist in F1 and F2, while the signals at 1473 and 1609 cm^−1^, observable in F1 and F2, cannot be seen in F3 and the paper. The stretching band between 3400–3300 cm^−1^ region gets lower in intensity in F1, F2, as it occurs in the spectrum of the dry gel. The spectrum of point F1, which has the highest dye density, is the most similar to the spectrum of the dry gel, for instance, the peak at 784 cm^−1^ is only observable in these spectra. Finally, peaks at 840 and 896 cm^−1^, which are not present in point F3 and the paper, are observed in the fingerprint region of the spectra acquired in correspondence to both the point F1 and the dry gel area. These differences in spectra of F1 and F2 are attributed to the photographic emulsion transferred on paper. In particular, the signals at 830–840 and 890–897 cm^−1^ are reported in the transmission spectra of hydroxyethyl cellulose, with reference to [57]. Moreover, the single firm peak at around 1140–1145 cm^−1^ found in the spectra of the dry gel, F1, and F2 is also present in the hydroxyethyl cellulose. The comparison with the FTIR reflection spectra of hydroxyethyl cellulose (directly on the powder and on the film obtained from the drying of its dispersion in water) confirmed this attribution (see Appendix A), even if it is fundamental to highlight that some artifacts are present in the reference spectra due to several factors. In particular, if the peak at 893 cm^−1^ is observable in all the spectra, it is important to mention a probable overlapping with a close band at 899 cm^−1^ of the paper. The band at 835 cm^−1^ could be affected from deformations for the aggregation state of the material: it results barely visible in the solid-state spectrum, while it has a maximum in the spectrum of the film from the water dispersion. The band at 1140 cm^−1^, instead, presents a derivative-like shape in both the reference spectra, while it could overlap with paper bands in the mock-up spectrum. Considering the materials used for producing Polaroid gels [47], where the viscosity of the reagent was provided by water-soluble polymeric thickeners including hydroxyethyl cellulose, the attribution of the above-mentioned peaks to hydroxyethyl cellulose should be cited taking into account these spectral deformations and the eventual overlapping with cellulose signals.

### 2.2. Paolo Gioli’s Original Artwork

#### 2.2.1. FORS

The paper of Gioli’s artwork has an absorption maximum at 374 nm, which is found to be identical to the paper which is used for the mock-up.

All points turn out to have maxima in similar wavelengths to those of yellow, magenta, and cyan dyes used in the Polacolor 2 system, which also correspond to our FORS results from the mock-up emulsion transfer (Table 3). However, in the original artwork of Paolo Gioli, the colours are much better preserved simply because the artist used recently expired or non-expired films. Therefore, the maxima of the bands obtained from the FORS analysis result in different intensities based on the prevailing dye found in the point of interest, and the band intensities are significantly higher compared to the mock-up (Figure 6).

With reference to the maxima of the bands reported in Table 3, hypotheses about the molecular structure of the dyes can be formulated. The wide maximum at 450 nm, evident in G1 and G4, can be attributed to the yellow dye used in Polacolor 89, a chromium (III) complex of azomethine type dye showing a maximum at this wavelength as expected from a Polacor 2 dye complex [48,49]. The absorption density is higher in the zone of yellow and red (400–600 nm) and lower in in the zone of blue (600–700 nm) which explains the reddish colour of the point.

The maxima around 535 and 575 nm, which change in relative intensity according to the overlapping with the bands of other dyes, are indicative of the magenta dye. These maxima differ from the analogue observed at 567 nm for the mock-up, so a different dye can be hypothesized. The difference is probably caused by the X substituent group of the magenta dye complex, which is fundamental for the differentiation of the dye complex. In the original artwork of Gioli, Y group is C_6_H_5_, the Z group is CH_3_, and the X group is found to be a cyanide group (CN) with reference to [35] (Figure 7).

A main band at 680 nm, along with another one at about 615–620 nm, is observable in all the points, even if its intensity is dominant for G5. These signals, as observed for the mock-up, are compatible with the characteristic ones of copper phthalocyanine compounds [15,24,48,49]. 

#### 2.2.2. FTIR

The spectra of the points GF3 and GF4 had interference fringes that made it difficult to acquire accurate information. Therefore, the results obtained from points GF3 and GF4 were not taken into consideration. The main signals for the spectra of points GF1, GF2, GF5, and paper are shown in Table 4. 

It can be stated that all three spectra are significantly similar to each other even though they represent different areas of the artwork having different hues (Figure 8). The characteristic signals of cellulose, found in the paper and reported in Section 2.1.2, can be noted [56]. 

Compared with FTIR spectra of the the mock-up emulsion transfer, the spectra of Paolo Gioli’s original artwork shows major similarities. In the fingerprint region of the three spectra, all points present peak around 825–830, 893–904, and 1139–1145 cm^−1^ which show their analogues in the spectrum obtained for the dry gel. As observed in the mock-up spectra, these peaks could be indicative of hydroxyethyl cellulose, even if a certain overlapping with the signals of cellulose cannot be excluded, as discussed above, which would explain a certain shift (Appendix A). With reference to the literature, in a study that covers the identification of copper complexes of pyrazolone dyes by FTIR in transmission mode [54], the signals in the range at 1600–1680 cm^−1^ are due to the aromatic functionality of the pyrazolone dyes. The absorption bands around 1445–1465 cm^−1^ may indicate the presence of N=N stretching vibrations. It is fundamental to highlight that these signals overlap with those of cellulose at 1641 and 1433 cm^−1^, respectively, but the higher similarity between GF1 and GF2 spectra and the dry gel one in comparison to the paper one is evident: in particular for GF1, the spectrum shows the same pattern in the 1580–1680 cm^−1^, with two maxima at 1604 and 1655 cm^−1^, observable for the dry gel and attributable to aromatic groups of pyrazolone dye, while no maximum for the cellulose is evident at 1641 cm^−1^ [58]. These signals would confirm the presence of pyrazolone complexes, constituting the magenta and yellow dyes, previously considered based on FORS data.

## 3. Materials and Methods

### 3.1. Sample Preparation

In order to test the diagnostic protocol, a Polaroid transfer mock-up was prepared following a technique similar to Paolo Gioli’s. The following equipment has been used: Polaroid Polacolor Type 88; receptor paper: Fabriano water colour papers (as a smoother surface will capture the image better, hot pressed watercolour paper was preferred); Polaroid EE 100 land camera; a rolling pin; a hard smooth working surface; scissors; timer; surgical gloves. The dry transfer method consisted of the steps explained in detail by [59].

The chosen artwork of Paolo Gioli to be analysed takes part of the “Cameron Obscura” series that he created in 1981.

### 3.2. The Spectroscopic Analysis

For the analysis of the mock-up created by transferring the Polacolor 88 film to paper and the original artwork of Paolo Gioli, three different spectroscopic methods are chosen to be applied: FORS (Fiber Optics Reflectance Spectroscopy, Plainsboro, NJ, USA), Raman Spectroscopy and FTIR (Fourier Transform InfraRed, Ettlingen, Germany) spectroscopy. The reason to apply more than one analytical method is from the simple fact that each method is complementary to each other, and therefore, their results are comparable and more informative [60,61,62].

#### 3.2.1. FORS

##### Analysis of the Mock-Up Polaroid Emulsion Transfer

In this research, FORS was used to have a preliminary and complementary method to determine the dyes and the dye developers found in the transferred Polaroid 88 emulsion transfer on paper. 

Since the film was expired and out of the developing gel sacs, of the eight films placed in the camera, only one was in a liquid phase, and only one mock-up could be obtained. The developing gel contained in the other seven films was dry and therefore could not provide an image. The film containing the liquid gel had two sacks, and only one has broken to release the gel. Therefore, the image obtained had an irregular form, though it presented different tonalities of a similar hue. For statistical purposes, 20 points were analysed in correspondence with the overall image on the mock-up, while an area was analysed on the gel contained in the sac (Appendix A). For every analysed point, 10 spectra were acquired. 

The analysis was conducted using Exemplar LS from B&W Tek, Inc., Plainsboro, NJ, USA, with a wavelength Range of 200 to 850 nm, spectral Resolution 1.5 nm together with the BPS101 Tungsten Halogen Light Source with a spectral range of e 350 nm to > 2600 nm. Data were acquired via a handheld probe, with an analysis area of 2 mm^2^, placed at 45° in close contact with the sample. This equipment was calibrated using a certified reflectance standard provided by B&W Tek, Inc., Plainsboro, NJ, USA. Spectra were recorded in 10 scans with an integration time of 80 µs par scan. In order to compare the obtained spectra with absorbance ones available in literature for dyes, we calculated the apparent absorbance spectra from the reflectance ones using Origin 2021 Software, according to procedures reported in literature [52,54,55] and using the conversion in Equation (1):(1)$$\mathrm{Abs}\propto \log(\frac{1}{R})$$
where Abs is the absorbance and *R* is the reflection spectrum. The experimental apparent absorption spectra can be used for qualitative identification of the dyes based on the apparent absorption maxima, assuming that transmission of the light through the sample can be considered neglectable. It has been preferred to analyse mainly the absorption spectra of the dyes because spectral data of the dyes in the literature are mainly found in solutions and, therefore, in absorbance. Comparing dyes in a solid state with the dyes in solution involves some limitations to their identification, since differences among spectra of the same molecule in different phases must be considerd (different wavelenght maxima, presence of multiple bands, different band broadness, etc.) However, taking into account the scarce availability of standards and reference spectra of solids for this typology of analytes and with reference to the fact that this is a preliminary study for Polaroid matrices, this approach was chosen.

##### Analysis of Paolo Gioli’s Original Artwork

The analysis was conducted with the same instrumentation as the mock-up sample. Spectra were recorded with 70 µs integration time with 10 acquisitions. Inflection points in the reflectance mode were determined using the first derivative spectrum, and the pseudo absorbance spectrum was generated by calculating the negative logarithm of the reflectance of the sample of each spectrum using Origin 2021 Software. Ten points, including the paper, were chosen to be analysed from Gioli’s original artwork (see Appendix A). Ten acquisitions were made from each point.

#### 3.2.2. Raman

Three different points were chosen from the transferred emulsion on paper sample, R1 being the darkest, R2 being the medium, and R3 being the lightest in hue (see Appendix A). Paper and the dry gel contained in the sac of the Polacolor 88 were also analysed.

The instrument used was LabRAM HR Evolution Confocal Raman Microscope (laser wavelength: 633 nm), by HORIBA, Kyoto, Japan. The used objective was 100× for each point. For the paper 100% laser intensity was set, for the dry gel 2.5%, and for the transferred emulsion on paper 5%. For all points, the acquisition ranges were 100–1000, 1000–1900, and 2400–3300 cm^−1^. The acquisition time for paper was 20 s with 30 acquisitions, for the dry gel 3 s with 500 acquisitions, and for the transferred emulsion on paper, it was 5 s with 60 acquisitions.

However, the results obtained from Raman Spectroscopy did not provide information about the analysed sample. In fact, strong interference fringes were observed. For this reason, the method was not used for the Paolo Gioli’s artwork. Indeed, with reference to the great potential of Raman spectroscopy in providing molecular information, this analytical issue should be deepened. This will be the object of further studies, when other experimental conditions will be tested: for instance, variation of the spot size. However, we preferred highlighting this aspect in order to make the reader aware of possible setbacks for other studies on similar matrices.

#### 3.2.3. FTIR

##### Analysis of the Mock-Up Sample

The instrumentation used for the FTIR analysis was Bruker Optics-IFS 66 v/s Vacuum FT-IR, Ettlingen, Germany. The instrument was combined with the Hyperion 2000 IR microscope with 15× objective and used in the Reflection mode, since the sample was not transparent enough to transmit the IR radiation. Combining microscopy with FTIR allowed us to perform a completely non-invasive analysis.

Four different points were chosen from the transferred emulsion on paper sample, F1 being the darkest, F2 being the medium, and F3 being the lightest in hue and P0 being the paper (see Appendix A). In addition, the powdered dry developing gel was analysed too. As reference, a small gold plate as a reflective surface was used. Three measurements were taken from every sample point using the following parameters: 4 cm^−1^ resolution, 256 scans, wavenumber covering from 400 to 6000 cm^−1^ using OPUS software. To obtain a pseudo-absorbance spectrum, the following Equation (2) is used (*R* = reflectance, *R*_1_ = reference:(2)$$\mathrm{Abs}\propto -\log(\frac{R}{R_{1}})$$

##### Analysis of Paolo Gioli’s Original Artwork

The instrumentation used for the FTIR was the same as the mock-up. Five different points were chosen from the transferred emulsion on paper sample (see Appendix A). As reference, a small gold plate as a reflective surface was used. Three measurements were taken from every sample point using the following parameters: 4 cm^−1^ resolution, 256 scans, wavenumber covering 400 to 6000 cm^−1^ using OPUS software. A pseudo absorbance spectrum was obtained with the above-mentioned equation.

## 4. Conclusions

The aim of this research was the characterization of the Polaroid emulsion transfer in Paolo Gioli’s artworks, contributing preliminary knowledge to the literature about the materials of the artworks and the appropriate protocols which can be applied for future studies. We faced several limitations during the making of the mock-up sample due to the aging of the Polaroid 88 film. The film expired in November 1985. Therefore, the alkaline gel, which was supposed to reach the film layers to provide a positive image, was dry. Moreover, since production of these films was discontinued by the Polaroid Corporation, it was significantly hard to find the Polacolor films that Gioli used. 

When performing FTIR, combination of the microscope with the FTIR in reflectance mode was used, since our sample will not be transparent enough to transmit IR radiation and since it was crucial to perform a technique that was non-invasive. The obtained spectra from FTIR gave significant results about the cellulose found in the paper, and the presence of Arabic gum as a binder of the paper was detected. FTIR results also showed the presence of hydroxyethyl cellulose as the thickening agent of the gel contained in the sac (in our case in a dry state), differing from the fingerprint region of only the paper itself. However, this assignation is only hypothetical because working in reflectance mode brought limitations to the assignment of the peaks due to spectral deformation and overlapping with cellulose signals. The FTIR spectra acquired from the original artwork of Paolo Gioli showed very similar results to our mock-up sample, but the presence of pyrazolone dyes could be confirmed from the spectra of the points analysed. Furthermore, the presence of hydroxyethyl cellulose as the thickening agent of the gel was evaluated also in this case. However, further research is necessary to define the efficacy of FTIR for the analysis of Polaroid emulsions transferred on paper by Paolo Gioli. In particular, extended studies on the materials present in Polacolor film, aimed to precise individuation of spectral artifacts and determination of single component contribution to the spectrum, are required, and they will be the object of further studies. About the determination of the dye developers present in both the mock-up sample, obtained from Polacolor 88, and Paolo Gioli’s Original artwork created with Polacolor type 89, the most diagnostic results were obtained from FORS. With reference to the literature, the main dyes used in the Polacolor Types 88 and 89 were determined from the apparent absorbance maxima, which allowed hypothesizing their structures and molecular substituents. However, other studies involving further characterization of original materials are necessary, in order to confirm the data reported in literature and also to support these results. The characterization of the dye developers with complementary techniques (e.g., chromatography, mass spectrometry [63,64]) is foreseen and desirable. In conclusion, new perspectives are open: from the conservation science point of view, Polaroid film characterization represents a new frontier to be investigated, whose results could be fundamental for future preservation, as it could be the case with Paolo Gioli’s works. Further research, involving complementary analytical techniques and ageing studies, is required in order to deepen our knowledge about instant photography materials, conservation, and occurrences.

## Figures and Tables

**Figure 1 molecules-27-07023-f001:**
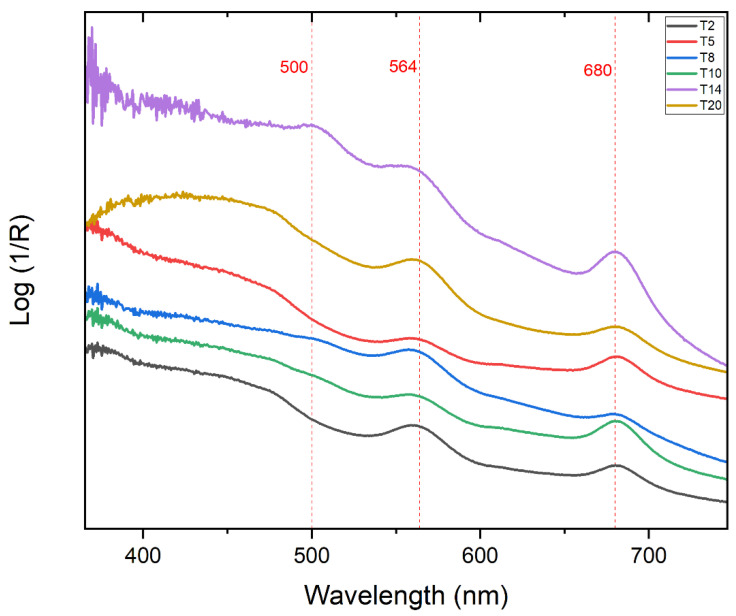
Absorption bands of the sample Polaroid 88 transferred on paper obtained by FORS spectrophotometer.

**Figure 2 molecules-27-07023-f002:**
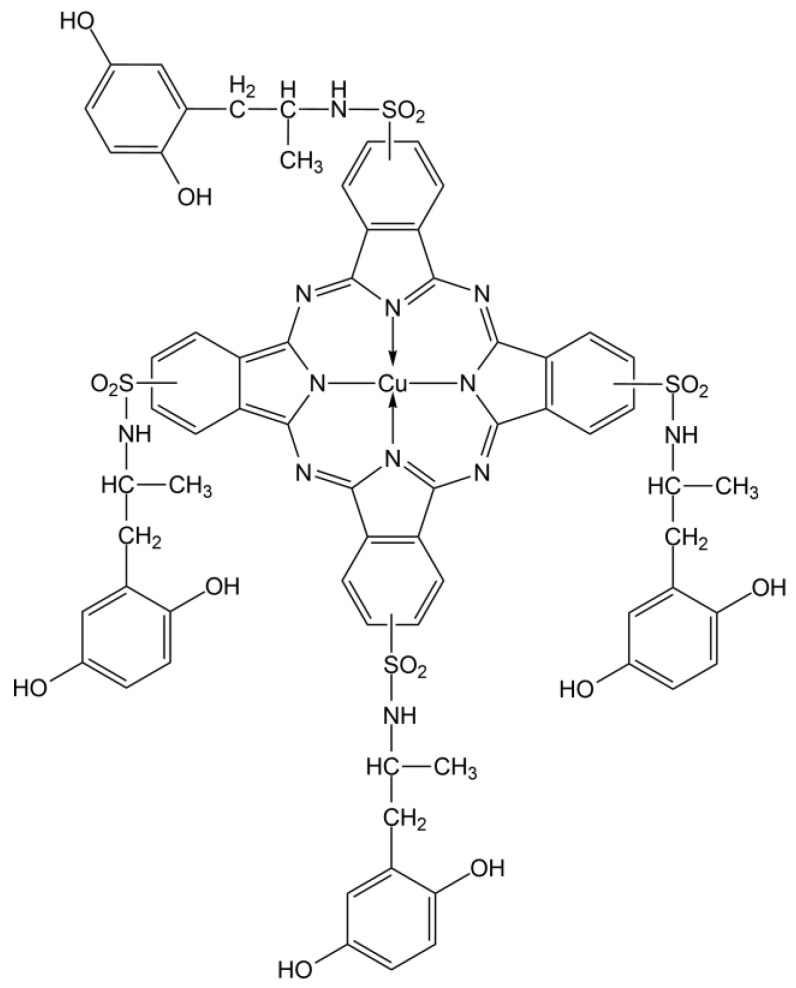
Blue copper phthalocyanine compound.

**Figure 3 molecules-27-07023-f003:**
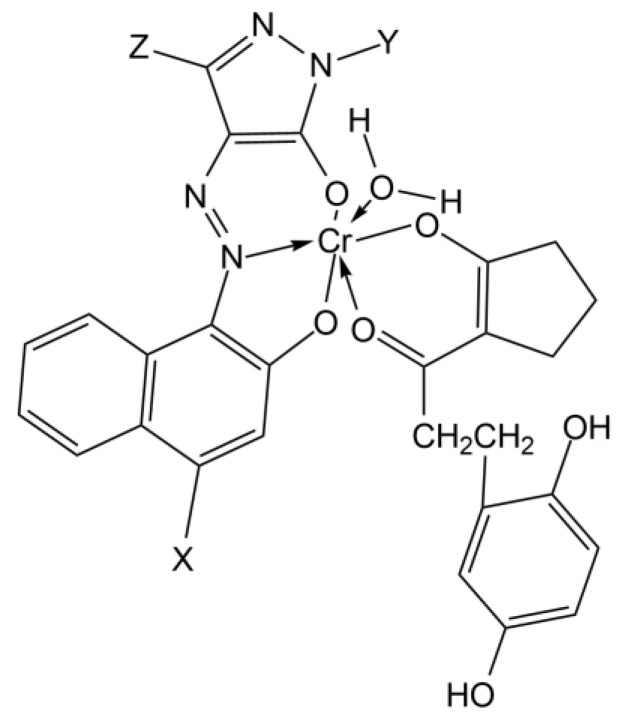
The general structure for a Chromium complexed magenta dye developer in Polaroid films; the X, Y, and Z substituents could vary in different dyes of the same typology.

**Figure 4 molecules-27-07023-f004:**
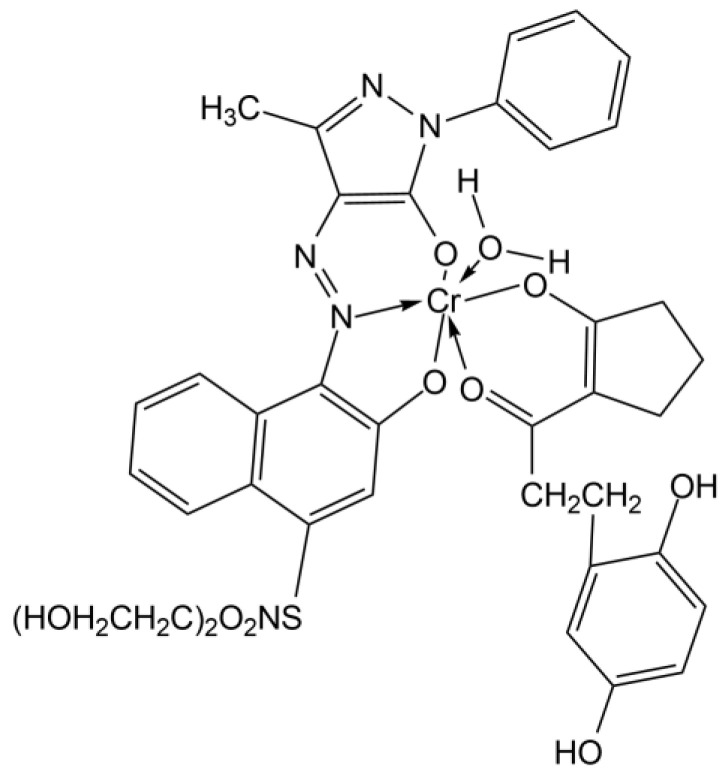
Structure of the magenta dye developer present in Polacolor 2 films and identified in the studied mock-up.

**Figure 5 molecules-27-07023-f005:**
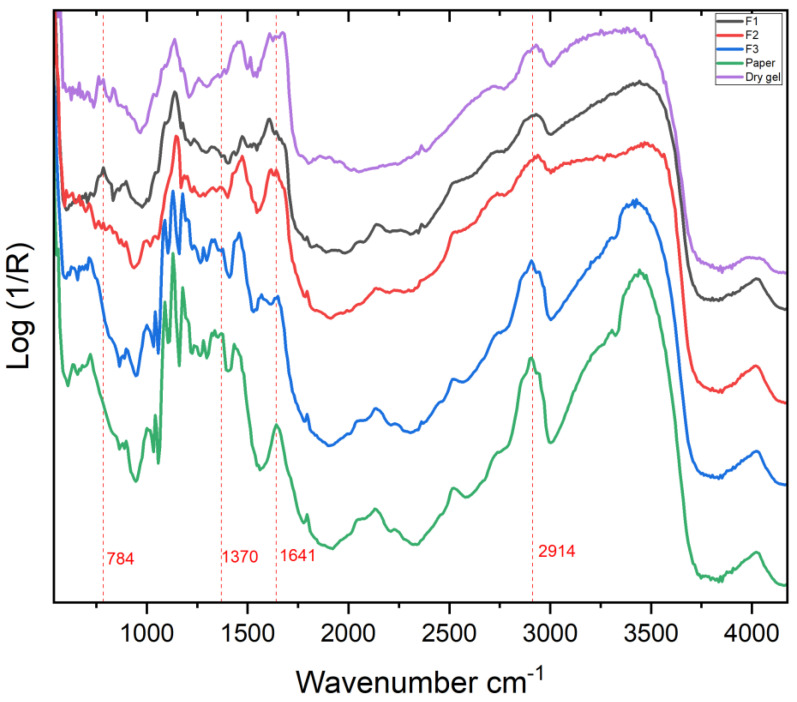
FTIR spectra of points F1, F2, F3, paper, and the dry obtained from the mock-up sample.

**Figure 6 molecules-27-07023-f006:**
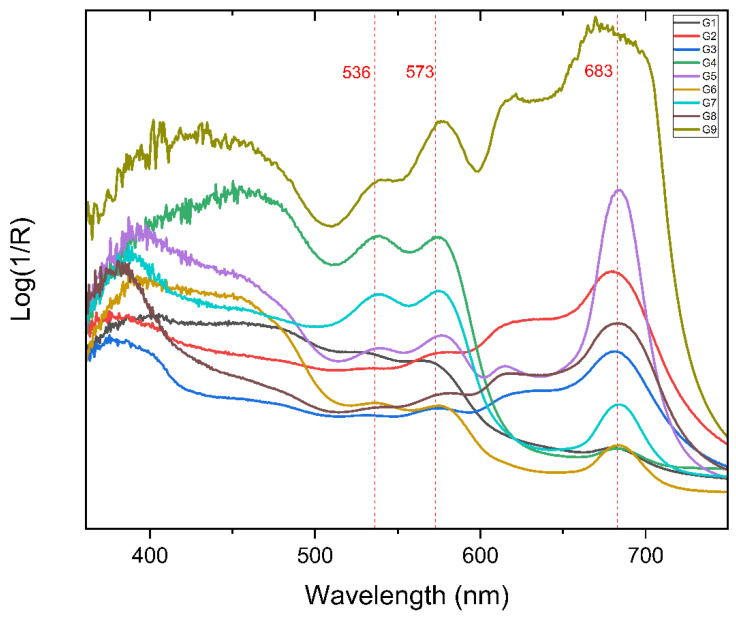
Absorption bands obtained by FORS spectrophotometer from Paolo Gioli’s original artwork.

**Figure 7 molecules-27-07023-f007:**
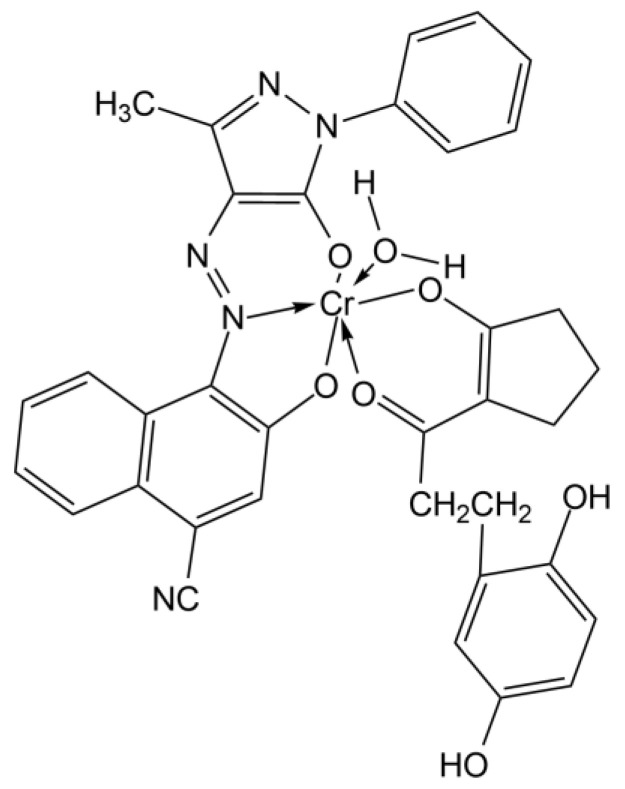
Magenta a chromium complex of an azo pyrazolone of type.

**Figure 8 molecules-27-07023-f008:**
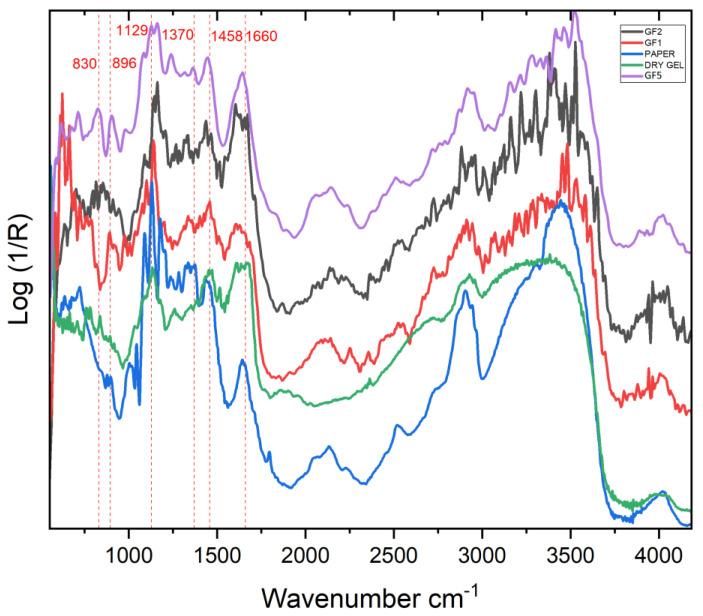
FTIR spectra of points GF1, GF2, and GF5; of the paper obtained from the original artwork of Gioli; and of the reference dry gel. Signals of the dry gel are highlighted.

**Table 1 molecules-27-07023-t001:** Absorption maxima of the mock-up Polaroid 88 transferred on paper obtained by FORS spectrophotometer.

Points	Absorbance Max (nm)
T1	560, 682
T2	564, 685
T3	564, 683
T4	500, 561, 682
T5	443, 564, 682
T6	501, 564, 682
T7	500, 564, 685
T8	562, 682
T9	505, 562, 682
T10	560, 682
T11	501, 565, 682
T12	502, 563, 682
T13	501, 560, 682
T14	503, 564, 682
T15	505, 564, 684
T16	563, 684
T17	564, 683
T18	505, 564,683
T19	562, 682
T20	456, 564, 684

**Table 2 molecules-27-07023-t002:** The L* a* b*data of the acquired spectra of sample points T1–T20.

Point	L*	a*	b*
T1	78.37	7.243	23.830
T2	68.49	4.805	19.961
T3	65.46	6.279	18.444
T4	59.29	6.329	14.921
T5	65.13	3.359	23.101
T6	49.90	8.333	14.516
T7	45.78	7.725	13.677
T8	54.82	10.898	13.000
T9	39.94	9.091	12.545
T10	63.73	6.769	18.838
T11	41.12	13.288	15.784
T12	47.02	11.160	15.140
T13	46.17	11.040	15.779
T14	30.58	13.827	13.844
T15	40.39	11.700	14.025
T16	58.35	7.935	17.138
T17	54.93	7.852	15.279
T18	49.30	15.432	14.817
T19	52.27	10.981	16.194
T20	56.07	12.205	21.171

**Table 3 molecules-27-07023-t003:** Absorption maxima of the points chosen from Paolo Gioli’s original artwork obtained by FORS spectrophotometer.

Points	Absorbance Max (nm)
G1	450, 535, 569, 681
G2	570, 620, 680
G3	528, 570, 626, 680
G4	450, 536, 573, 684
G5	536, 573, 615, 684
G6	536, 575, 683
G7	536, 575, 683
G8	580, 615, 683
G9	577, 617, 677

**Table 4 molecules-27-07023-t004:** Bands of the FTIR spectra of points GF1, GF2, G5, and paper.

Acquisition Point	Wavenumber
GF1	724, 820, 893, 976, 1078, 1099, 1139, 1206, 1313, 1335, 1386, 1403, 1459, 1481, 1604, 1655, 2109, 2725, 2911, 3482, 4004, 5141 cm^−1^
GF2	820, 904, 945, 1066, 1088, 1145, 1161, 1206, 1268, 1285, 1313, 1335, 1352, 1380, 1436, 1464, 1492, 1604, 1655, 1671, 2103, 2137, 2523, 2725, 2882, 3409, 4026, 4777, 5153 cm^−1^
G5	713, 825, 903, 1083, 1127, 1162, 1240, 1318, 1364, 1448, 1644, 2103, 2143, 2513, 2720, 2866, 2916, 3522, 4021, 4766, 5203 cm^−1^
Paper	720, 877, 899, 1003, 1016, 1043, 1090, 1130, 1176, 1205, 1236, 1282, 1321, 1336, 1357, 1373, 1433, 1452, 1643, 1795, 2069, 2129, 2518, 2723, 2904, 3443, 4021, 4749, 5177 cm^−1^

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
