# Peer review of "Photons for Photography: A First Diagnostic Approach to Polaroid Emulsion Transfer on Paper in Paolo Gioli’s Artworks"

_molecules, 2022, doi:10.3390/molecules27207023_

Round 1
Reviewer 1 Report
The authors are applying three spectroscopic methods in order to analyze mock-ups and an original artwork of Paolo Gioli.
The authors do state their intention of developing an analytical protocol, along with provide insights into the material used by the artist.
The authors reached somehow a part of their goals.
On the development of analytical protocols to be used in future studies, as the authors are stating in the conclusion, there was minimal achievement of this.
Raman spectroscopy proved to be of very little used due to high fluorescence, I assume. The authors do give a very speculative explanation which is had to grasp. But they do still list Raman as a method and having it in two chapter which I find misleading given there are no relevant result. This could have been merely as a sentence in the conclusion part.
The FTIR analysis do show mainly bands for paper or and other cellulose based materials. This is not of any surprise and it is not understandable why the authors use a whole chapter (3.1.3) to describe very detailed the bands belonging to cellulose and the shifts of the signals. Which is the purpose of it? Also, the authors keep repeating within the same section, the same information several times. This is utterly unnecessary, makes the reading very difficult and bring nothing news. Chapters 3.1.3 and 3.2.2 are nearly identical with sentences with look like copy pasted. " From the FTIR results it can be stated that the area where the emulsion transfer is made shows mostly the paper than the dye itself".
To my understanding, the authors compared IR reflectance spectra with transmission spectra. The reflectance spectra of solids with transmission of liquids (at least this one understands from the way authors express themselves). Both are scientifically incorrect.
The authors do attribute though some back from the IR spectra to the dyes: 1660-1600 cm-1 for carbonyl and 1458 cm-1 for N=N vibrations. How about the very strong contribution of the carbonyl groups from cellulose? Is the band from 1458 completely absent in the paper spectrum? These issues have to be addressed given the strong interference of the paper, fact also acknowledged by the authors.
FORS method seems to be the one fitting the best the purpose of the paper and, indeed, the authors do insist very much on its results. But, again, they do repeat themselves making the discussion unnecessarily long. For example, it was mentioned three different times (3.1.1) that the paper signals do no interfere with the ones of the dyes. This is again repeated in 3.2.1.
The discussion of the FORS is extremely descriptive and it could be reduced by half. I strongly recommend the authors to do that.
The colorimetric data appear from nowhere in the FORS part, but only for the mockups and the discussion is very difficult to understand. Why was this done only for mock-ups and not for the artwork itself? Table 2 is also incorrect.
In 2.1.1, it is stated that "For statistical purposes, ten spectra were acquired from 20 points" - not really clear how can 10 spectra be recorded from 20 points? Was it 20 spectra per 10 points? 2 spectra per point?
Apart of these shortcomings, there are also typing mistakes and grammar mistakes. The style is heavy and the language is stretched.
Along with redesigning the article (including the title), a language wash is utterly necessary.
Reviewer 2 Report
Nice paper with some really clearly thought-out structure. I'm not convinced the molecule models are necessary or helpful nor add anything substantial to the paper, not least as they seem at times to be speculative. That said, it is a molecules publication and they add eye-candy.
For this stage in review there should be line numbers so I will instead quote more text to locate required changes.
Intro:
must pay well - should read - must pay good attention
Polo - should read - Paulo
References start at 3! What happened to 1 & 2?
James Clerk - should read - James Clerk Maxwell (I'm related to him)
Corporation Afterwards - should read Corporation. Afterwards
hight viscosity - should read - high viscosity
hight molecular weight - should read - high molecular weight
further light and stability - should read - further light fastness and stability
Section 2.2.1.1:
80000 ns - surely we can use 80 µs keep SI prefixes in scale
intervals of 10 - what does this mean (I use spectrometers every day and if its not clear to me then its not clear)
equation 1 (and 2)- why? Are you trying to give optics people a heart attack?
I understand why you state this but it needs a little more explanation around it and please at least say proportional to rather than equal to.
70000 ns as above
q.v. 2.2.2 and 2.2.1 you swap between wavelength and wavenumber, perhaps neatest to add in wavenumbers in the FORS in addition to WL to keep tidy.
2.2.3.2 Fully reflective? at best 98.5%R
Figure 1,5,8 no Y-axis (this is fine) but why then add to figure 6?
Points T4,T6 etc- after this why not refer to CIE coordinates ? The wavelengths, illuminants etc are well known
Table 2 first line says table 1 - which is it and renumber!
3.1.2
Raman is awesome, use a smaller spot size and this would work well for you
As know from the literature - should read - as known...
out mock-up sample - should read - our mock..
3.2.1
colorant - should read -colourant (or be consistent)
page 13 bottom dye dye - should read - dye
page 14 bottom also are also - should read - are also
Otherwise very good - top job.
Round 2
Reviewer 1 Report
The paper gained some clarity, but the authors seem to totally overlook the fact the paper is way too descriptive and too much wording was used. Instead of cutting from the wording, the authors added even more text.
A lot of the data included in the text could be actually presented in a table: this would make the text easier to read and the flow better.
And again, repetitions must be avoided.
At page 5, "The most representative bands are found to be belonging to the points T2, T5, T8, 48 T14, T10, T20 since they show the most distinct and characteristic bands (Figure 1)." This sentence has no logical meaning.
At page 6, in the text before fig 1, the author should honestly add about the limitations comparing spectra of solid with the ones of solutions.
Page 7, paragraph 7-8: what do the authors mean by "the absorption of red"?
Page 8: is figure 4 supposed to be according to the description from page 7? It is not. Moreover, the general formula of Chromium complexed dye from fi 3 differs from the one of fig 4. Is it correct?
Page 8, paragraphs 11-15: the authors do speak about ageing and degradation like they are two different processes. Is it so?
Page 9, paragraph 2: the authors mention the spectrum of T1 from Fig 1, but there is no T1 spectrum over there. Moreover, the wide curve 409-480 nm is really difficult to observe in T2 spectrum.
Page 9, par 20-21: what kind of "similar results"?
Page 9, par 27: "a more similar band to the paper" or a more similar spectrum?
Regarding the merely tentative attribution: hydroxyethyl cellulose is readily commercially available, and given the authors do posses the instrumentation, why not purchasing hydroxyethyl cellulose and producing a spectrum? That would be so helpful for other scientists.
Page 17, figure 8: there is no spectrum GF5 over there!
The authors should include a table containing the which measured points belong to the mock-ups and which to the artwork. Or, better, a picture. Otherwise is really difficult to read the text.
Author Response
See the attachments.
